

# Analyzing the proximity to cover in a landscape of fear: a new approach applied to fine-scale habitat use by rabbits facing feral cat predation on Kerguelen archipelago

Pierrick Blanchard[1], Christine Lauzeral[1], Simon Chamaillé-Jammes[2], Nigel G. Yoccoz[3] and Dominique Pontier[4]

[1] Université de Toulouse, Université Toulouse III Paul Sabatier, CNRS, ENFA, UMR 5174 (Laboratoire Évolution et Diversité Biologique), Toulouse, France
[2] Centre d'Écologie Fonctionnelle et Évolutive, CNRS, UMR 5175, Montpellier, France
[3] Department of Arctic and Marine Biology, The Arctic University of Norway, Tromsø, Norway
[4] Université de Lyon, Université Lyon I Claude Bernard, CNRS, UMR 5558 LBBE (Laboratoire Biométrie et Biologie Evolutive), Villeurbanne, France

Corresponding author
Pierrick Blanchard,
pierrick.blanchard@univ-tlse3.fr

## ABSTRACT

Although proximity to cover has been routinely considered as an explanatory variable in studies investigating prey behavioral adjustments to predation pressure, the way it shapes risk perception still remains equivocal. This paradox arises from both the ambivalent nature of cover as potentially both obstructive and protective, making its impact on risk perception complex and context-dependent, and from the choice of the proxy used to measure proximity to cover in the field, which leads to an incomplete picture of the landscape of fear experienced by the prey. Here, we study a simple predator-prey-habitat system, i.e., rabbits *Oryctolagus cuniculus* facing feral cat *Felis catus* predation on Kerguelen archipelago. We assess how cover shapes risk perception in prey and develop an easily implementable field method to improve the estimation of proximity to cover. In contrast to protocols considering the "distance to nearest cover", we focus on the overall "area to cover". We show that fine-scale habitat use by rabbits is clearly related to our measure, in accordance with our hypothesis of higher risk in patches with smaller area to cover in this predator-prey-habitat system. In contrast, classical measures of proximity to cover are not retained in the best predictive models of habitat use. The use of this new approach, together with a more in-depth consideration of contrasting properties of cover, could help to better understand the role of this complex yet decisive parameter for predator-prey ecology.

## INTRODUCTION

Cover, hereafter defined as any tangible feature in the habitat that impairs the prey's and/or the predator's ability to see and/or move (i.e., the definition of "structural cover" by *Mysterud & Østbye (1999)*, restricted to the "landscape of fear" context of *Laundré,*

*Hernández & Ripple (2010)*), has often been considered as an explanatory variable in field studies investigating prey behavioral adjustments to predation risk (*Caro, 2005*). Yet, the way cover shapes risk perception in prey species and their subsequent antipredator tactics still remains equivocal (e.g., *Burger, Safina & Gochfeld, 2000*; *Tchabovsky et al., 2001*; *Caro, 2005*). This may be due to the ambiguity inherent to the use of a single word to refer to multiscale habitat objects (e.g., a prey may have its visual field impaired by a tree line hundreds of meters away and by a tuft of vegetation near its eyes when feeding head down) or to the paucity of studies considering simultaneously these different scales (e.g., *Pays et al., 2012*). Two additional reasons may explain why the role of this parameter remains ambivalent.

First, cover is a (visual/physical) barrier for the focal individual prey but also for its predator(s). Hence, the ratio between its contrasting obstructive (i.e., prevents the prey from seeing or escaping the predator) and protective (i.e., prevents the predator from seeing or attacking the prey) properties (*Lazarus & Symonds, 1992*; *Mysterud & Østbye, 1999*) and thus the overall risk perception is highly specific to a predator(s)-prey system. This obstructive/protective ratio depends on the intrinsic physical properties of the cover itself (e.g., dimensions, opacity), in relation to characteristics such as the body size, sensory capabilities, and hunting/escape tactics of particular predator and prey species (e.g., *Lima, 1990*; *Murray et al., 1995*; *Newberry & Shackleton, 1997*). Furthermore, intraspecific variability is expected in the above described characteristics and also in other traits (e.g., sex, reproductive status, experience or group size) that may also determine risk perception in relation to a specific cover type (e.g., *Götmark & Hohlfält, 1995*; *Bowyer et al., 1999*; *Stratmann & Taborsky, 2014*; *Beauchamp, 2014*). Finally, time of day (e.g., *Moreno, Delibes & Villafuerte, 1996*) or season (e.g., *Bowyer et al., 1999*) may affect the obstructive/protective ratio for a given cover type and individual. Hence, the contrasting results of studies investigating the role of cover in shaping risk perception, even those performed on the same species and type of cover (e.g., *Jaksic & Soriguer, 1981*; *Moreno, Delibes & Villafuerte, 1996*), may reflect natural heterogeneity.

The second reason why the role of cover for prey remains ambivalent is that field measurements are often too basic to provide biologically relevant proxies. For any given time and predator-prey system, the obstructive/protective ratio of cover may vary with proximity. For instance, cover provided by drooping branches has a low obstructive/protective ratio when a prey is close because the prey is less visible, but its ability to detect a predator and flee is unaffected. As distance from the branches increases, the ratio increases because the prey becomes more visible and the branches provide a possible ambush site for a predator. Hence, field studies classically consider the "distance to cover" (when mentioned, typically the "nearest" "principal" cover) as a routine measurement (*Caro, 2005*). We speculate that part of the variability in the results of studies relating prey behavioral traits to cover is a consequence of using a single distance measurement that leads to an incomplete picture of the landscape of fear experienced by the prey. The problem with distance to nearest cover is that this measure ignores the dimension of the edge of the cover the prey faces and the presence of additional cover. Yet, in a case of an overall obstructive cover, a forest edge of 100 m long provides more potential

ambush sites than a patch of trees of 10 m long even if the shortest distance between the cover and the prey is the same in both situations. Similarly, a prey surrounded by several shrubs is expected to be at higher risk than if there is a single shrub in its vicinity in case of obstructive cover. Hence, although the shortest distance to cover is of importance as this gives an indication of the shortest time lag before being predated (or sheltered in case of overall protective cover), it is only part of the information. There is thus a need for a measure of all distances between the prey position and cover in the surroundings (see also *Metcalfe, 1984*; *Gómez-Serrano & López-López, 2014*), i.e., a need for an area to cover. We suggest that such a metric would provide a more reliable measure of the proximity to cover and thus, of risk perception, than the commonly used "distance to the nearest cover." In the present paper, our aims were to (1) develop such a metric, from field measurements to their geometrical analysis and to (2) use this metric to investigate habitat use of rabbits (*Oryctolagus cuniculus*) facing predation threat by feral cats (*Felis catus*) on the Kerguelen subantarctic archipelago. We also considered classical proxies of proximity to cover in order to allow comparisons.

In studies of responses to cover, there is a risk of circularity if the property of cover is inferred from its effect on behavior (*Lazarus & Symonds, 1992*). Instead, cover property, and thus the way it is expected to trigger risk perception in prey, should arise from the knowledge of the system. Hence, making a prediction requires assessing the risk perception (i.e., the obstructive/protective ratio) inherent to the (different types of) cover(s), i.e., being able to relate the physical characteristics of the cover such as opacity and size to the escape tactics of the prey and the hunting tactics of the predator involved. Furthermore, it is important to recognize that cover may be associated with foraging profitability (e.g., *Morgantini & Hudson, 1985*), for example if it directly provides food or affects food plants through shading or nutrients (e.g., *Mysterud & Østbye, 1999*; *Dellafiore et al., 2014*). In the present paper, we took advantage of a simple predator-prey-habitat system allowing us to hypothesize that predation threat was the main driving force of habitat use by rabbits. Given the specific characteristics of the system, we predicted that rabbits should avoid patches with high proximity to cover, i.e., with small area to cover.

## MATERIALS AND METHODS

### Study site

Introduced by sailors during the nineteenth century, rabbits are now widespread throughout the Kerguelen archipelago (*Chapuis, Boussès & Barnaud, 1994*). Domestic cats were introduced in 1951 to control invasive rodents (*Rattus rattus, Mus musculus*) and rabbits at the research station of Port-aux-Français. Cats are now widely distributed over the main island (Grande Terre), where the study took place (Pointe Morne area, 49°22′S, 70°26′E).

Our study was performed in December 2014. We focused on a ca. 100 by 700 m area covered by mounds less than 2 m high, formed of earth and roots and covered by the perennial herb *Acaena magellanica* (Rosaceae) (see File S1). The space between the mounds was composed of *Acaena magellanica*, *Poa annua* and bare ground/rocks. The

study area is surrounded by open meadows with flat topography, covered with dense swards of *Acaena magellanica*.

We censused active rabbit burrows as indicated by fresh pellets or evidence of animal passage in a 0.73 km$^2$ area including the patches (see below) used in this study. We found 51 active burrows, all outside our 70,000 m$^2$ study area. Although we do not know the relationship between the number of active burrows and the population size in this habitat, previous published relationships suggest about 22 rabbits (i.e., about 30 rabbits/km$^2$) (*Ballinger & Morgan, 2002*). Fifteen additional active burrows were present outside the 0.73 km$^2$ area, 800 m away from the study area.

## Predicting the effect of area to cover on habitat use by rabbits

The following characteristics of our system allowed us to reliably assess the role of cover in shaping risk perception in rabbits.

— Food resources. We selected patches of a single preferred plant species, *Poa annua* (*Chapuis, Boussès & Barnaud, 1994*). This highly nutritious alien grass represents most of rabbit diet in our study area (over 90% of the plant fragments found in fecal pellets at the time of the year our study took place, *Boussès, Arthur & Chapuis, 1988*). As the study area is relatively restricted, meteorological and edaphic conditions are very similar. Moreover, *Poa annua* was heavily grazed (1–2 cm high) throughout the study area. Because patches were likely to be similar in food quality and quantity, the effect of cover was not confounded by foraging profitability. Finally, rabbits face no interspecific competition for feeding resources in this habitat (in particular, reindeer *Rangifer tarandus* have not been observed in the study area).

— Predators. Predation by brown skua (*Catharacta lonnbergi*) on rabbits occurs on the Kerguelen archipelago, but mostly on small islands (*Chapuis, Boussès & Barnaud, 1994*) and on young/sick rabbits (the myxomatosis virus was introduced in 1950's to control populations). Given that our study site was on the main island, that no skuas nested nearby, that no rabbits were observed or killed (as part of other protocols) with apparent signs of myxomatosis, that our study took place before the rabbit birth period, that cats were observed daily in our study area and finally that rabbits are the primary prey of cats in Kerguelen archipelago (*Pontier et al., 2002*), we believe that predation pressure experienced by rabbits in this habitat is mostly due to cats. This contrasts with other studies on rabbits and other prey where predators are often diverse. Our field observations of hunting bouts revealed that cats are stalk-and-ambush predators (although they also visit burrows). A foraging rabbit is clearly at risk if surprised by a cat, while early visual detection of the cat allows escape, especially in open areas.

— Cover types. We focused on a habitat with a single type of cover, earth mounds (i.e., visually opaque and physically impenetrable, see File S1), and considered as "cover object" any mound higher than 20 cm (i.e., capable of hiding an ambushing cat from a rabbit, even in an upright posture, and of hiding a rabbit from a cat, unless the rabbit was in an upright posture). Most mounds were taller than 1 m. Cats may attack directly from a side of a mound, but also from behind a smaller mound or possibly from the top of a larger mound.

The characteristics of this predator-prey-habitat system allowed us to consider cover as a source of risk for rabbits, i.e., far more obstructive (total opacity in a context of stalk-and-ambush predator threat and complete physical barrier when escaping, with no intrinsic refuge property–once potential burrows (see below) are statistically accounted for) than protective (rabbits hidden from cats). Accordingly, we predicted that rabbits would favor patches with larger areas to cover, i.e., with larger unobstructed areas.

## "Patch" characterization and data collection

We defined a "patch" as a circular area with a 2 m diameter, covered exclusively with *Poa annua*, whose center was at least 20 m away from the center of another patch. The studied area was fully searched for patches, which numbered 32.

We used total dry weight of pellets per patch as a measure of patch use. In every patch, we made a one-time collection of all the fecal pellets, thereby assuming that the disappearance time of pellets was not related to the variables we considered. We also kept the pellet collection time very short (2 days) to reduce the occurrence of additional pellets during the study period. Pellets were subsequently dried for 4 days at 40 °C (i.e., until their weight stopped decreasing) and weighed. Pellet quantity is a reliable method to assess rabbit abundance at the habitat scale (*Palomares & Delibes, 1997*; *Palomares, 2001*; *Cabrera-Rodriguez, 2006*). At a finer scale, pellet quantity has been shown to index the frequency of rabbit visitation of the patches (*Bakker et al., 2005*).

In every patch, a single observer took the following measurements:

—The GPS coordinates. This allowed us subsequently to investigate statistically the spatial structure of our response variable, the pellet total dry weight.

—The total number of old burrows within a 20 m of diameter circle around the center of the patch. This parameter was recorded because we observed that old burrows represented potential refuges. We considered the hypothesis that there were more rabbits closer to active burrows (all localized outside our study area, see above) as part of our investigation of spatial structure. The number of burrows is classically part of rabbit habitat selection studies (e.g., *Palomares & Delibes, 1997*).

— The terrestrial distance (m) to the nearest old burrow, bypassing a mound if present.

— The number of "contact points" with *Poa annua* around the focal patch. Contact points were located 1, 3 and 5 m from the center of the focal patch every 45° (i.e., $n = 24$ in total for each patch). A proxy of the isolation of the focal patch of *Poa annua* was then calculated as the frequency of "contact points" without *Poa annua*. We included this parameter because we hypothesized that the attraction of a patch could have been positively related to how isolated it was from other patches of *Poa annua* or conversely, that patches surrounded by a high overall *Poa annua/Acaena* ratio could have been more attractive for rabbits. Moreover, this measure allowed us to investigate the hypothesis that risk perception would be increased if abundant *Acaena* around the focal patch impaired the visual field of a rabbit feeding head down.

— The distances (m) from the center of the patch to the nearest mound and to the nearest side of a mound visible from the patch, within a biologically relevant area (see below). In addition to their biological relevance as an obstacle when escaping or a point

from which a hidden cat may attack, these distances allow us to compare our proposed measure with other measures of risk.

— The total number of mound sides visible from the patch inside the biologically relevant area (see below) and the mean distance (m) to these sides.

— Additionally, we recorded the unobstructed area for each patch. The description of this measure is presented below.

## Measuring the unobstructed area

From the center of the patch, the observer scanned 360° using a rangefinder with angle display (Vector 1500 GMD). Each time the rectilinear edge of a mound (i.e., forming a straight line as a whole) started and stopped, the distance from the center of the patch and the corresponding angle were recorded. When a mound edge did not appear to be rectilinear, the measurements were recorded for each of its rectilinear segments. This gave us a set of triangles with an angle and the length of the two adjacent sides, allowing the calculation of their area. However, the sum of these areas would provide a poor proxy of the unobstructed area experienced by a rabbit because some mounds would be too far away to provide a cat with a successful ambush position or to provide a physical barrier to escape. Furthermore, considering such distant mounds may inflate the unobstructed area, potentially masking biologically relevant effects at shorter distances. We thus calculated the unobstructed area within a circle. Considering such a circle also allowed us to deal with cases where no cover occurred before the horizon (a single sector in a single patch for our study area). We set the circle radius based on our field observations of hunting behavior by cats. Because the longest cat run toward a feeding rabbit that we observed was about 25 m, we first considered this distance. Then, in order to identify the radius of the circle with the highest predictive power, we computed the squared coefficient of determination between the observed and the fitted values for models built with values of unobstructed areas calculated for circles with radiuses ranging from 1 to 150 m (with 1 m increments).

Depending on whether the rectilinear edge of the mound the observer faced fell entirely inside the circle, entirely outside the circle or was secant in one or two points, we used different formulas to calculate the corresponding area, as explained Figs. 1A–1C (see also the script used to compute the unobstructed area, written in the R language and provided in File S2). The sum of these areas provides the unobstructed areas, ranging from 25.3 to 1646.1 $m^2$ for the circle with a radius of 25 m.

## Statistical analyses

The pellet total dry weight exhibited significant positive autocorrelation (Moran's $I = 0.111$, $p < 0.001$, Fig. 2). We used generalized least squares (GLS) models to account for autocorrelation in model residuals (*Selmi & Boulinier, 2001*). Different models of spatial structure (assuming spherical, exponential and Gaussian structures) were fitted and the best fitting model (exponential in all the cases) was defined using the Akaike Information Criterion (AIC; *Selmi & Boulinier, 2001*; *Diniz-Filho, Bini & Hawkins, 2003*).

We log transformed the pellet total dry weight to meet the assumptions of constant variance and normality of the residuals. To avoid collinearity issues, we only considered

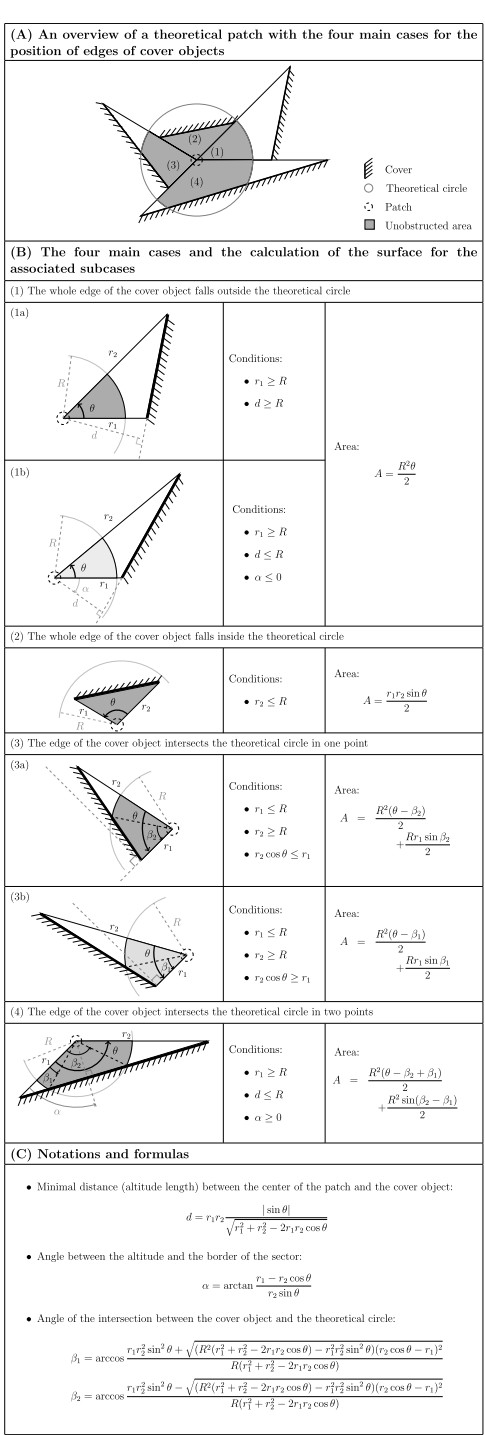

**Figure 1** **The different potential cases for the position of edges of cover objects in relation to a circle around the patch and the corresponding formulas for the calculation of the area ("A" in (B)) between the center of the patch and the cover object or the circle.** The unobstructed area is calculated by summing all these areas (see also File S2). Cases numbered (1) to (4) in (A) refer to the same numbers in (B). Dark grey areas in (B) refer to subcases illustrated in (A); light grey is used for alternative subcases. Inequalities in (B) are not strict: for limiting cases, the different corresponding formulas lead to the same results. All the angles are counterclockwise. $R$ is the radius of the circle. $d$, $\alpha$, $\beta_1$ and $\beta_2$ are defined in (C). We defined $r_1 \leq r_2$. The only required field measurements are $r_1$, $r_2$ and $\theta$.

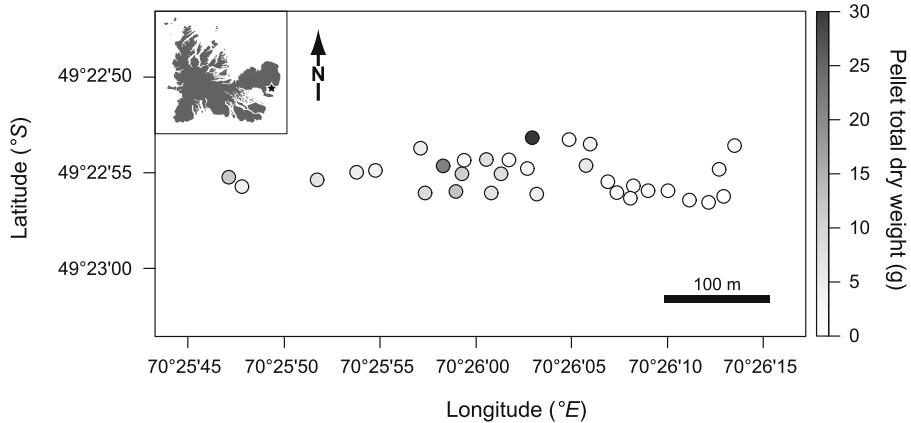

**Figure 2** Pellet total dry weight (indexed by grey levels shown on the right $y$-axis) measured in the studied patches (dots) according to their spatial position (left $y$-axis and $x$-axis).

models including explanatory variables that were not significantly correlated (all $p >$ 0.14). We did not include interactions among explanatory variables because of the small sample size. For patches with no burrows within a 20 m of diameter circle ($n = 7$), the variable "distance to nearest burrow" was missing. Hence, we first tested the effect of this variable on a subsample and then reran the models without it to avoid artificially reducing sample size when testing the other explanatory variables. We proceeded in the same way for "distance to nearest side" ($n = 2$ patches with no visible side) and for "mean distance to sides" (same 2 patches). We selected the final model by fitting the complete model and removing each term successively. The significance of each term was determined by assessing the change in deviance (i.e., Likelihood Ratio Test–LRT) against a chi$^2$ distribution with the appropriate degrees of freedom. For nonsignificant variables considered in several models, we present the maximum LR value, the corresponding minimum $p$-value and the estimates. Estimates were all computed on standardized variables (mean $= 0$, S.D. $= 1$) to allow comparisons of effect sizes not dependent on measurement scale (*Gelman & Hill, 2007*). Analyses were performed in R 3.1.2 (*R Core Team, 2014*) using the packages *ape* for spatial analyses and *nlme* for developing the models. The French Polar Research Institute approved this program (number 279).

## RESULTS

The unobstructed area within a 25 m radius circle around the patch positively affected the pellet total dry weight ($df = 1$, LR $= 9.264$, $p = 0.002$; estimates: intercept $= 1.41 \pm 0.20$ S.E., slope $= 0.51 \pm 0.16$ S.E.; Fig. 3). When the unobstructed area increased from 500 to 1,000 m$^2$, the predicted pellet total dry weight increased from 3.38 to 7.02 g.

The mean distance to visible mound sides and, to a lesser extent, the distance to the nearest visible side were also positively related to the pellet total dry weight (mean distance to mound sides: $df = 1$, LR $= 4.721$, $p = 0.030$; estimates: intercept $= 1.40 \pm 0.23$ S.E., slope $= 0.39 \pm 0.17$ S.E.; distance to the nearest side: $df = 1$, LR $= 3.776$, $p = 0.052$; estimates: intercept $= 1.39 \pm 0.24$ S.E., slope $= 0.35 \pm 0.18$ S.E.). Finally, the total number

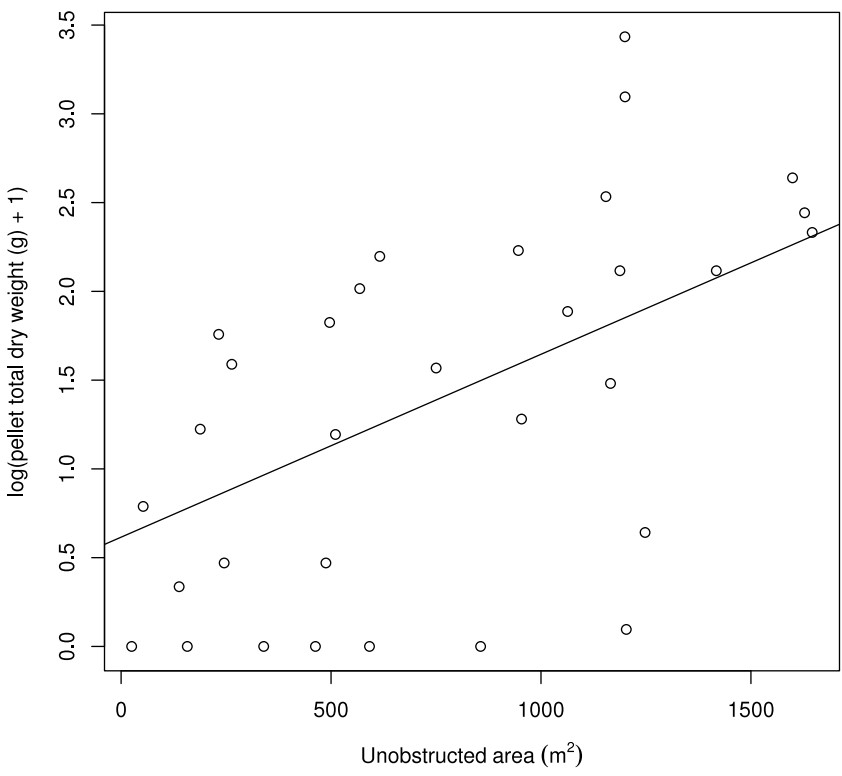

**Figure 3** **The positive relationship between the log(pellet total dry weight (g) + 1) and the unobstructed area (m²) computed for a circle with a 25 m radius.**

of visible sides within a 25 m circle negatively affected the pellet total dry weight ($df = 1$, LR $= 5.032$, $p = 0.025$; estimates: intercept $= 1.42 \pm 0.27$ S.E., slope $= -0.37 \pm 0.16$ S.E.).

The other explanatory variables, including the distance to the nearest mound, were not retained in the final models and had smaller effect sizes as measured by standardized regression coefficients (distance to the nearest mound: $df = 1$, LR $= 2.156$, $p = 0.142$; estimates: intercept $= 1.44 \pm 0.27$ S.E., slope $= 0.25 \pm 0.17$ S.E.; total number of old burrows: $df = 1$, LR $= 0.247$, $p = 0.619$; estimates: intercept $= 1.40 \pm 0.19$ S.E., slope $= -0.08 \pm 0.14$ S.E.; distance to the nearest old burrow: $df = 1$, LR $= 1.847$, $p = 0.174$; estimates: intercept $= 1.27 \pm 0.51$ S.E., slope $= -0.27 \pm 0.18$ S.E. ; isolation: $df = 1$, LR $= 1.014$, $p = 0.314$; estimates: intercept $= 1.41 \pm 0.31$ S.E., slope $= -0.15 \pm 0.15$ S.E.).

The hypothesis that the proximity to active burrows partly explains the spatial pattern we report (Fig. 2) might be relevant as active burrows were localized at about 100 m northwest from the patches, i.e., on the side of the highest values in pellet total dry weight.

Finally, the best radius of the circle ranged between 19 and 32 m (based on maximizing the coefficient of determination between the observed and the fitted values), in line with our field observations of cat hunting behaviors (Fig. 4).

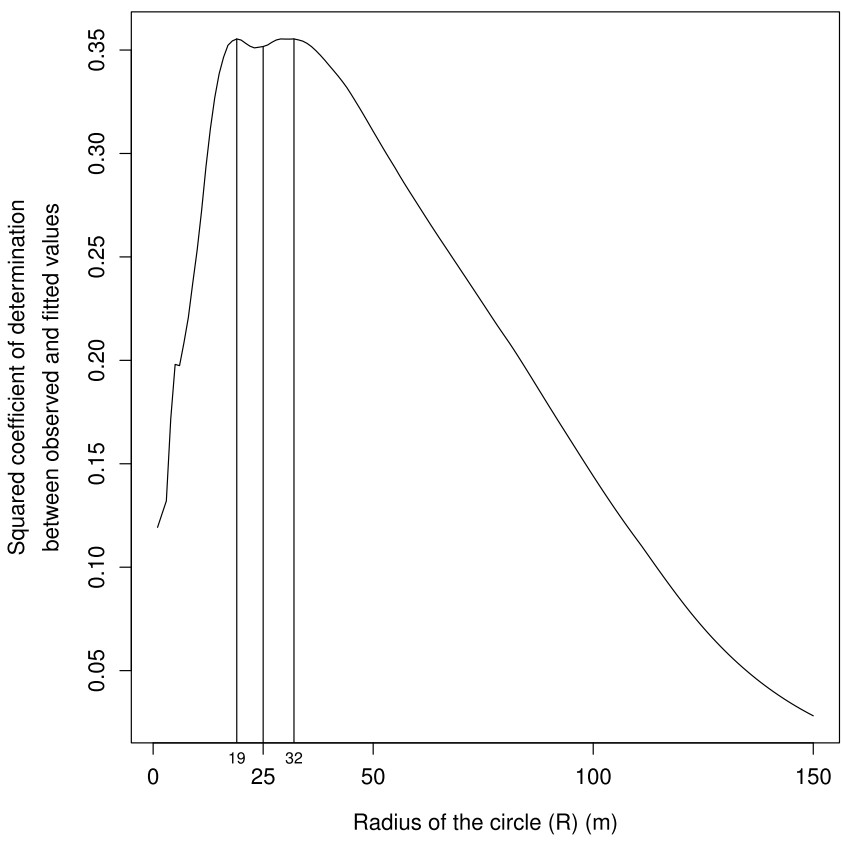

**Figure 4** **Squared coefficients of determination between the observed and the fitted values for models built with values of unobstructed areas calculated for circles of radiuses (R in Fig. 1, in m) ranging from 1 to 150 m and plotted on *x* axis.** The biologically relevant distance ranges between 19 and 32 m, in line with our field observations of cat hunting bouts.

## DISCUSSION

We took advantage of a study area with patches composed of a single preferred plant species, displaying cover objects of a single type (i.e., a single intrinsic obstructive/protective ratio) and where predation risk arose from only one species: feral cats. In this context, our results strongly suggest that cats shape fine-scale habitat use by rabbits. Fewer pellets were present in patches with smaller unobstructed areas, i.e., with smaller visible area from the center of a 25 m radius circle around the patch, and thus closer potential danger, and with greater proximity of physical barrier, and thus reduced escape possibilities.

The consideration of the area to cover helped to understand the role of cover in fine-scale habitat use by rabbits. With classical approaches, i.e., with the distance to the "nearest cover" (i.e., nearest mound or nearest visible side and thus shortest time to initiate an escape and shortest distance before reaching a barrier to escape/an exit) as an explanatory variable, the conclusion of an absence of (or weak) differences between the patches in the amount of pellets according to the proximity of cover would have emerged (i.e., no or weak effect of nearest mound or nearest visible side distance in our analyses,

respectively). Yet, our results strongly suggest that proximity to cover shapes habitat use by rabbits in this habitat.

Our method requires no particular expertise–we provide an R-script to calculate the required areas (see File S2). Although a rangefinder with angle displays may be expensive, this can be replaced by a basic model without angle displays (or even by a tape measure) and a compass. A limitation of our method is the subjectivity in classifying a portion of the cover object as rectilinear or not. However, this subjectivity is more pronounced when the cover is far, i.e., when the rabbit also faces visual limitations.

Beside the role of predation risk as a driver of the pattern we report, we cannot rule out the influence of nonexclusive additional pressures. Reduced visibility may also lead to decreased opportunity to monitor conspecifics while foraging. This may again indirectly relate to predation risk because of the concomitant loss of information about conspecifics' vigilance/escape behavior and because of the decreased "confusion effect" (*Beauchamp, 2014*), as other prey individuals are also less visible to an attacking predator. However, reduced visibility is further expected to decrease foraging and social opportunities (e.g., localization of high quality patches, scrounging, gathering information about potential mates/competitors) (*Beauchamp, 2014* for a review, *Monclús & Rödel, 2008* for rabbits). Smaller unobstructed area is also expected to mechanistically lead to lower overall surface of edible plants (i.e., more mounds and thus less accessible plants per unit area) and thus to increased foraging effort/gram of ingested grass/unit time, i.e., to possible decreased attractiveness. Finally, we cannot rule out the possibility that patches with small unobstructed areas are more difficult to find for rabbits.

Pellet weight suggests that rabbits spend more time in patches with larger unobstructed areas. However, the similar height of grass suggests that different patches are foraged at similar rates which would be expected to take similar amounts of time. Several hypotheses may explain this apparent contradiction. First, because we did not measure grass height, we cannot rule out the possibility that patches with larger unobstructed areas had slightly shorter grasses as a result of more time spent foraging. Second, although we believe meteorological and edaphic conditions are very similar over our restricted study area, we cannot rule out the possibility of fine-scale variations. This could lead to higher *Poa annua* productivity in some patches, and thus to increased foraging time despite similar height of grass. For instance, patches with larger unobstructed areas may benefit from increased sunlight. Finally, under the hypothesis of similar foraging rates among patches, behavioral explanations may be proposed. Rabbits may increase grazing rate (through bite rate/size) and thus decrease exposure time in patches with smaller unobstructed areas. Additional activities such as grooming, resting or playing may also occur in safer places (e.g., *Cowlishaw, 1997*; *Blumstein, 1998*), leading to an increase in the overall amount of time spent.

## Past and future studies

The use of an incomplete measure of the proximity to cover may explain the absence of significant results in studies investigating behavioral adjustments of prey in relation to cover (e.g., *Pays, Ekori & Fritz, 2014*) while other studies in the same population

report such an effect (e.g., *Périquet et al., 2012*), depending on which particular cover object is considered. Investigating the predictive power of our proxy in these prey-predators-habitats systems, including large-scale cover objects such as forest edges, may be informative. Previous studies on habitat selection by rabbits in relation to cover reported contrasting results, possibly reflecting the ambivalent properties of cover and the large range of predators rabbits face in their worldwide distribution (*Courchamp, Chapuis & Pascal, 2003*). While the importance of the protective function of cover in rabbit habitat selection has been shown by several studies (*Villafuerte & Moreno, 1997*; *Dellafiore et al., 2014*), leading to a greater use of patches farther away from cover when predation pressure is lower (*Banks, Hume & Crowe, 1999*), cover may also be avoided when it reduces the visual field (*Moreno, Delibes & Villafuerte, 1996*). Part of the variability among studies may further be explained by the use of the distance to nearest cover, widespread in this species also (e.g., *Moreno, Delibes & Villafuerte, 1996*; *Villafuerte & Moreno, 1997*). Moreover, the obstructive/protective ratio of a given cover for a given population may vary according to the period of the day (*Moreno, Delibes & Villafuerte, 1996*): rabbits have been reported to preferentially feed closer to cover during the day (hiding from birds of prey) than at night (avoiding stalking carnivorous mammals).

Cover presenting total visual and physical obstruction, as in our study, is probably uncommon. Assessing the obstructive/protective ratio and thus the risk perception associated with a cover may require consideration of its visual and physical properties (*Schooley, Sharpe & Van Horne, 1996*; *Camp et al., 2012*). Moreover, it is probably uncommon to have only a single type of cover as in our study. The "area to cover" should thus be calculated separately for each cover type. Cover objects can all be of the same nature (e.g., protective/obstructive properties > 1) but differ in the relative intensity of protective and obstructive properties, or can be of opposite nature (e.g., some with protective/obstructive properties > 1 and some others with a ratio < 1). For example, a feeding patch for a mountain ungulate in the vicinity of a cliff and of several shrubs and rocks may be characterized by the overall area to the shrubs and rocks (with a risk increased for patches with smaller areas) and by the overall area to the cliff (with a risk decreased for patches with smaller areas) if the individuals face stalk-and-ambush predators unable to reach their prey on a cliff.

## ACKNOWLEDGEMENTS

We thank Fabien Egal, Johan Chervaux and Adrien Tavernier for field assistance, Gaël Grenouillet for help with spatial analyses, two anonymous referees and editor D. L. Kramer for very helpful comments on earlier drafts.

### Funding

Logistical and financial support were provided by the French Polar Institute (IPEV, program number 279) and the Zone Atelier "Recherches sur l'Environnement Antarctique

et Subanctarctique." PB and CL are part of the French Laboratory of Excellence project "TULIP" (ANR-10-LABX-41; ANR-11-IDEX-0002-02). The funders had no role in study design, data collection and analysis, decision to publish, or preparation of the manuscript.

## Grant Disclosures

The following grant information was disclosed by the authors:
French Polar Institute (279).
Zone Atelier Recherches sur l'Environnement Antarctique et Subanctarctique.
TULIP: ANR-10-LABX-41, ANR-11-IDEX-0002-02.

## Competing Interests

Nigel G. Yoccoz is an Academic Editor for PeerJ.

## Author Contributions

- Pierrick Blanchard conceived and designed the experiments, performed the experiments, analyzed the data, contributed reagents/materials/analysis tools, wrote the paper, prepared figures and/or tables.
- Christine Lauzeral contributed reagents/materials/analysis tools, prepared figures and/or tables, reviewed drafts of the paper.
- Simon Chamaillé-Jammes reviewed drafts of the paper, wrote the initial project for funding.
- Nigel G. Yoccoz analyzed the data, reviewed drafts of the paper.
- Dominique Pontier reviewed drafts of the paper, wrote the initial project for funding and supervises the program.

## Ethics

The following information was supplied relating to ethical approvals (i.e., approving body and any reference numbers):
    French Polar Institute (IPEV), program no 279.

## Data Availability

    The raw data has been provided in the Supplemental Information.

## Supplemental Information

Supplemental information for this article can be found online at http://dx.doi.org/10.7717/peerj.1769#supplemental-information.

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
