# Peer review of "Analyzing the proximity to cover in a landscape of fear: a new approach applied to fine-scale habitat use by rabbits facing feral cat predation on Kerguelen archipelago"

_PeerJ, doi:10.7717/peerj.1769_

## Round 0.1 · original submission · Major Revisions

This manuscript presents a method for calculating a domain of safety (or unobstructed area, as a reviewer suggests) around a potential foraging patch based on distances to obstructive cover. The authors show that their measure provides stronger predictive power for use of patches by rabbits (measured by fecal pellet mass) than alternative measures based on nearest cover. Both reviewers accept that the method provides a useful way to measure the potential impact of obstructive cover. However, one reviewer questions whether it would ever be likely that a simple distance to cover measure would be appropriate in the type of situation studied. Furthermore, both reviewers are concerned about the interpretation of the results because food availability is likely to be traded off with predation risk. One would expect less grass in the less risky patches that have more grazing and more fecal pellets. Because some individuals would be expected to forage in the riskier patch as a result of higher food availability, there will not be a 1:1 relationship between patch use and risk. This fundamental issue needs to be clarified. Both reviewers have indicated a number of other areas that need clarification.

Despite my familiarity with the topics involved, I found the manuscript quite difficult to read. This was in part due to long and rather convoluted explanations and in part to poor word choice. The Introduction and Discussion need to be focused more tightly on the actual contributions of your study. The manuscript requires the assistance of a fluent English speaker to improve the clarity of expression and word choice. I have submitted an annotated pdf of the manuscript with yellow highlights and inserted comments to indicate some (but far from all) of the cases where wording was awkward.

Reviewer 1 ·

Basic reporting

See General Comments

Experimental design

See General Comments

Validity of the findings

See General Comments

Additional comments

The main problem is the undefended assumption that the rabbits’ perception of risk can be assessed by ‘usage’ (here, measured as fecal pellet dry weight). This seems obvious, but it ain’t necessarily so. Both in theory and empirically, usage is also affected the amount or quality of the food available on each patch, which might vary. As just one example, the authors should consider a paper by Pomeroy (2005. Tradeoffs between food abundance and predation danger in spatial usage of a stopover site by western sandpipers, Calidris mauri . Oikos) which shows how food and danger interact to produce a non-intuitive pattern of usage.

Patches (line 181) and pellets line 187
In contrast with description of details of the domain of safety measures, this part is poorly explained. It seems that you defined as ‘patch’ a 2m diameter circle, but I have no sense for how large a portion of each sward this represents. Most of it? A small fraction? This might matter. I checked the Supplementary Information and here found a photograph, but it’s unlabeled, and I’m not sure what I’m looking at. It seems that the perennial herb Acaena magellanica surrounding the grass swards is perhaps 10 or 15 cm tall. If so, the edges of the patches might be perceived as more dangerous by rabbits, and if that is true the size of sward in which a patch is located might matter. Please expand on these points.

Pellets: How long a period of use does your collection represent? It isn’t clear if you cleared all the pellets away and then allowed them to accumulate again for a time, or whether you made a one-time collection. The former would be better.

Figure 2
‘Pellet total dry weight measured in the studied patches according to their spatial
Position’. The figure is just not very helpful. It could be improved by having a larger scale map of the study area. The legend doesn’t even say what the dots are, but I assume patch locations?

Figure 3 may be significant, but it is quite noisy (low variance explained). I don’t challenge the result, but in light of the comments about the importance of food, it might be worth considering where the variation comes from.

Reviewer 2 ·

Basic reporting

No comments

Experimental design

No comments

Validity of the findings

In this paper, the authors investigated how the structure of the landscape influences habitat use by rabbits in a simple environment including one major predator and one major resource type. Cover from small surrounding mounds was deemed obstructive, and the authors showed that the distance to the nearest mound proved a less reliable indicator of habitat use than a more complex index based on the surface free of obstruction around a patch used by rabbits. This paper adds to the debate about how to effectively measure what constitutes cover for an animal feeding in the open.

My first point is that I think it would have been very naïve to expect distance to nearest mound to have an effect on habitat use in this simple situation. The authors make a very long argument in the introduction about how simple measures based on distance to obstructive cover can miss the mark. I agree with these remarks, but does this really apply to the present results? Given that rabbits can be surrounded by mounds, I think what the authors investigated here is the effect of visual obstruction rather than the effect of distance to obstructive cover. When investigating distance to obstructive cover, it is quite clear usually that cover lies in one direction at a rather fixed distance from the focal animal. Obstructive cover in rabbits is much like rocks on a beach for foraging shorebirds (like in the study by Metcalfe (1984) on turnstones). In that study, measuring the distance to the nearest rock would have been meaningless. What matters here is to measure the degree of visual obstruction. I do not have a problem with the measurements taken in the present study. I just think there is a disconnect between the long introduction (and the long discussion) and the results. The results are simple and make a point about visual obstruction and its effect on habitat use. Do we really need all this discussion about properly assessing distance to cover?

My second point is about the choice of patches. The authors selected patches used by rabbits as judged by recent grazing. I would have simply measured patch characteristics in randomly selected areas. This way we could see whether there is a difference in habitat use across a much larger range of patches (grazed or not). Can you justify why you selected a different approach? Your approach also forces you to define a patch using an arbitrary distance of 20 m between two patch centroids.

My third point also relates to patch definition. As far as I can tell, the selected patches were grazed to the same extent. I would expect that to bring vegetation down to the same level in different areas would take the same number of rabbit-hours. Hence, I would expect the number of pellets to be the same in all the selected patches. What could explain differences in pellet numbers across these patches? I can imagine some patches were used by more rabbits or for longer, but given that there was no time window to start counting pellets (this is not mentioned in the paper but there was no beginning to pellet accumulation in a patch, just an end), would we not expect pellet numbers to become equal over time for the same level of exploitation? Is it possible that some of the pellets disappeared after a while so that only the fresh ones were counted? Is it possible that there is a non-linear relationship between pellet production and time spent in a patch (pellets start accumulating only after so much time in a patch but not before)?

My final point is about rabbit density. I did not get a good feeling for the number of rabbits present in the study area. Are we dealing with only 1 or 2 rabbits here? If this is the case, the study would be rather anecdotal. This information would also be useful to assess the relevance of alternative explanations put forward in the discussion (scrounging, social information, etc.). Are the rabbits territorial? This information again might help us assess the real sample size in the study in terms of rabbits. If the rabbits were territorial, would we not expect a negative relationship between ‘domain of safety’ and pellet numbers? The best patches would be used by fewer rabbits and so we would get fewer pellets.

Specific comments

Line 161: I can see that mounds are obstructive cover from the point of view of the rabbits, but I cannot really see how you can conclude that the cover is not also protective if the rabbits used it as cover from predators. Without actually watching predators and prey interact in the field, I would think it is rather hard to be certain about this.

Line 163: I think the term ‘domain of safety’ is not quite right. Why not simply call it the ‘unobstructed area’ surrounding a focal rabbit? This is objective and measurable.

Line 181: Can you specify why you selected these different patch characteristics? Are these related to rabbit biology?

Line 187: The study took place I assume over a period of 30 days. Is it possible that some of the pellets predate December? If so, do you have an idea how long pellets can last? This would help to address issue number 3. What about coprophagy? Can this affect pellet number differently in different types of patches?

Line 207: I was not quite clear what you meant by contact points.

Line 224: I was not sure I understood what you meant by a linear mound? Do you mean that the cover item had to form a straight line in the triangle?

Line 259: Why not simple drop this variable if it cannot be measured in nearly 25% of the patches? The number of burrows can be counted in each patch and can serve as a proxy for the distance to nearest burrow.

Line 328: Is this whole section really required? I would rather see a discussion of the results rather than a discussion of other methods.

---

## Round 0.2 · Minor Revisions

General comments

I apologize for the delay in responding to your revision. I decided that the manuscript was now clear enough that I did not need to send it out for another review. When I started to read it, I saw that many errors in English remained despite my request that you have a fluent English speaker review it. Instead of repeating my request, I thought I would able to point out the needed changes myself. As you can see from the attached annotated file, it turned out that errors were very numerous. Together with the holidays, this process substantially delayed completion of my review. I hope the improvements in the clarity of the manuscript justify the extra time.

In the annotated pdf, I note each error that I recognized by highlighting the problem areas in yellow and using attached comment to either propose an alternative wording for the highlighted area or another action such as 'delete'. The errors include spelling, grammar and incorrect word use. In some cases, the wording was not strictly wrong but I proposed alternatives that were in more common use or would be more easily understood. In quite a few cases, I proposed more concise alternatives to long and complex sentences. I hope that my suggestions are clear. If my wording changes your meaning, you can search for an alternative that addresses the problem while leaving the meaning intact.

You have repeatedly used an em dash where a hyphen should be used. I noted this problem in the Abstract, but you will have to correct it throughout the manuscript. For a description of correct use of hyphens and dashes, you could look at http://www.chicagomanualofstyle.org/qanda/data/faq/topics/HyphensEnDashesEmDashes/faq0002.html

I also inserted comments on the substance of your manuscript as I was reviewing it. Most of these are repeated, and sometimes elaborated, in my detailed comments below. I hope you will consider them carefully. You may contact me directly if any are unclear so that we can avoid another round of revisions. However, I consider these to be suggestions rather than requirements and that you as authors have the ultimate responsibility to choose how you present your findings.

Although my comments are numerous, they do not require any additional analyses or substantial reinterpretation, so I consider them to be 'minor revisions'.

Specific comments

Title
I think it would be useful to specify 'obstructive cover' in the title. Even though you want to keep the issue general, your contribution is strongly related to obstructive cover.

Abstract
L30 and elsewhere in the manuscript. "Distance to the closest cover item" is a rather awkward term and seems rarely used in the literature. In particular, 'item' is not often added to 'cover'. The term "distance to closest cover" would probably be more widely recognized by readers (as you wrote on L228). Throughout the manuscript, you refer to 'surface' when I think you should be specifying 'surface area'.

Introduction
L48. Meaning not clear: what is the interaction of multiple scales?
L69ff. This paragraph is very important to your Introduction, but I do not find it very clear. It raises multiple issues but fails to integrate them. The paragraph structure seems weak, so it is not easy to follow the logic using topic sentences. I tried to help by rewording particular sections, but I do not think I solved the larger problem. Consider the logical development of your ideas and the role of the topic sentences of paragraphs and revise accordingly. The sentence starting on L85 might make a better topic sentence.
L83. 'Ground cover' is a term used for low-growing herbs that cover the ground in areas where grasses do not grow, often in a horticultural context. I don't think that is what you mean here.
L89. This paragraph adds little to what has already been stated. I think you need to state why dimension is important in general. Even with your example, you treat the effect as obvious without actually explaining either the example or the general principle.
L93. Wrong term. I don't think you mean 'treeline' in the sense of the limit of habitat in which trees can grow in alpine or polar regions. Do you mean forest edge or a row of trees or something else? It is not clear why a line of trees would differ from a small patch.
L95. Again, you need to be explicit about why this is the case. It is not so intuitive that it needs no explanation, especially if thinking about protective cover. If you are trying to make a general argument, you need to consider both obstructive and protective effects. The effect of number of shrubs seems unlikely to have as strong an effect as a refuge as for potential ambush sites.
L97. This is a key justification of your study. Yet, it does not logically follow from the incomplete example of 1 vs. 2 shrubs. The logic needs to be developed. Also, this seems part of the same paragraph as (2) above.

Methods
L132. The approximately linear nature of the study area does not become apparent until the reader views Fig 2. I think it should be mentioned and perhaps explained here.
L133. Would it be useful to give a range and average height of the mounds, given the argument about cat attacks in relation to mound size? Also, what shape were the mounds? You refer to corners below, which implies that they were not round, yet the close mounds in the illustration do not seem to have obvious corners in the sense of a place where two edges meet.
L173. When you need to refer to a specific mound as cover, 'cover item' is acceptable. However, it seems awkward to me as compared to 'cover object'. As search of Web of Science suggested that neither appeared very often in abstracts of ecological articles but that object was somewhat more common (and nearly always for protective cover).
L182. As noted by the referee in the previous version, 'domain of safety' does not express well the concept involved. Although the referee does not explain why, it seems to me that this is because the animal is not equally safe throughout the area. It is at high risk near the mounds and lower risk at greater distances. I do not disagree that the area you measured should be positively correlated with safety and negatively correlated with risk, but this is not the concept conveyed by 'domain of safety/risk'. I do not see how the danger of circularity justifies your use of the term. It seems to me that the logic 'mounds produce visual and physical obstruction and little or no protection; therefore, we predict that use of an area by rabbits should be positively correlated with the unobstructed area' makes complete sense and is preferable to redefining unobstructed area as an area of safety without more information to identify the level of safety. Your second argument that different relative sizes or speeds might reverse the pattern seems to argue against 'area of safety' as much or more than it does against unobstructed area. I think the manuscript would be clearer if you replaced the sentences with something like: 'Accordingly, we predicted that rabbits would favor patches with larger unobstructed areas (or, alternatively, 'areas free of cover', if you prefer)'. Note the need to refer explicitly to 'area' when you refer to 'surface' because surfaces have other properties than area.
L191. Your Methods section implies that you measured all patches as defined rather than a random sample as stated in your response to the reviewers. Please clarify.
L193. Be careful not to switch between the terms pellets and droppings. This may confuse some readers, especially if not fluent in English.
L195. You also refer to pellet counts when you used dry weight. Perhaps counts are relevant to the literature (and are obviously well correlated) but you should be precise about what measures are involved in the various studies and use the appropriate term when referring to your own data.
L197. 'Visitation level' is not a generally recognized term. Please explain or choose a more general term.
L204/L132. The description of the study area and its relationship to active burrows is a bit confusing. Your study area measured 70,000 m sq. You counted 51 active burrows in an area about 10X larger and identified 15 additional burrows outside this larger area. Then you tell us that there were no fresh burrows in the study area. Assuming that 'fresh' the same as 'active', it seems odd that you provided an operational definition in the second use rather than first mention. Using two terms for the same concept in different places makes the reader more uncertain. It would have been useful to explain the absence of fresh burrows at the first reference because a reader could have reasonably inferred that some of the 51 were in the study area. At the second mention, you explain them as being known as part of another protocol, although you already mentioned it in your methods (if I understand correctly). How do these areas and burrows relate to Fig. 2? The panel frame seems to be much larger than 70,000 m sq but much smaller than 0.73 km sq and provides little help on identifying where the active burrows were because we can't determine where the border of the study area is so we can't judge where 800 m northwest would be.
L213. The only definition of nearest burrow I see above is for fresh burrows, but you seem to mean any burrow here. I left the reference to the definition out of my revised sentence, but clarify and include it if it is important.
L225. The description of distance to mound and corner is not clear. Do you mean that you only included mounds and corners within the fixed distance? What did you do if there were none? Did you do this for the whole range of potential distances? It seems to me that over this range you would have many more than the 7 cases of no mound or corner referred to in the statistical methods section.
L237. It is not clear how a mound forms a straight line. Are you referring to the edge where the mound meets flat ground or to the shape of the mound as a whole? How this concept would be applied to the circular mound in the foreground of your supplemental figure is not clear to me. Would this be considered rectilinear by ignoring the minor curve in the front? How did you deal with what appears to be the undercut nature of mounds - measure where it meets the ground or the maximal edge? I am not criticizing your method but simply trying to make sure that your method is understandable and repeatable.
L255. Does 'whether the mound fell entirely in the circle' refer to the position of the entire mound or to its rectilinear edge?
Fig. 1. I have provided numerous suggestions to clarify both Figure 1 and its caption on the annotated pdf.

Results
L309. Explain how you selected the range 19 - 29 m. It appears that the correlations are very similar for value outside this range, especially above 29 m.
Fig. 2. The statement about proximity to fresh burrows should be in the Results rather than the caption to be sure that readers understand the distinction between the effect of all burrows and the effect of more distant fresh/active burrows.
Figures 2, 3, and 4. Please see suggestions on the annotated pdf for clarifying captions and correcting grammar on axis labels.

Discussion
L340. Is the difficulty in classifying straight sections likely to make a large difference in the measure? My intuition is that the effect would be minor, but it would be worth explaining the importance of this if it could have a large effect.
L344. On L316, you wrote 'strongly suggests'. I'm not sure you have shown "beyond question" that predation is the driver. I agree that it is very likely, but the reasoning remains indirect. I suggest removing 'unquestionable', especially since you are arguing that there may be other benefits of avoiding sites with limited unobstructed area.
L345. I don't think that you can conclude that predation or the other factors is really a selective pressure in the evolutionary sense. Small-scale habitat selection could be the result of flexible behavior, not directly selection from specific environmental variables.
L354. I don't think 'overall time spent foraging' is the relevant variable here; it sounds like an individual time-budget variable. Furthermore, it is not completely evident that what you measured (essentially, a proxy for time spent per unit area in a grass-covered area) would be increased when more of the surrounding area covered with grass. In fact, with some assumptions, the opposite would be the case. The argument needs to be reconsidered and provided with more explanation to affirm its validity if it is correct.
L355ff. I think you need to be much more explicit about the potential contradiction in the topic sentence at the start of this paragraph to make sure that readers really understand. Although you somewhat addressed the reviewers' concerns here, it remains a bit obscure. You should have a topic sentence that indicates something along the lines of: 'Pellet weight suggests that rabbits spend more time in patches with larger domains of safety. However, the similar height of grass suggests that different patches are foraged at similar rates which would be expected to take similar amounts of time.' Think about how to most logically develop the ideas. The possibility that rabbits spend more non-foraging time (you mention grooming and play; what about rest?) in the less risky patches is one possibility. Is there any evidence from time time-budget studies that time spent in these activities could be large enough to make this difference? You have some sites in which pellet weight is zero. That can't easily be explained by not spending non-foraging time in riskier patches. Another possibility is that rabbits have a higher consumption rate of grass in riskier patches. Again, could the potential difference in foraging rate really account for the difference in apparent time spent (including sites that are supposedly grazed but totally lack fecal pellets)? You also raise the possibility that rabbits have shorter foraging bouts in riskier patches, but I don't see the logic of this if the total grazing time remains the same. There must be some assumption about defecation rate in long and short bouts that you are not making explicit. Is it also possible that there are differences in grass height among patches that you did not recognize because you did not actually measure grass height or that height of the grass is not determined by grazing?
L369. I don't see what these suggestions for other studies add to your contribution except a lot of references. I suggested deleting and focusing the paragraph on the discrepancy between the fecal indicator of time budget and the apparent lack of difference in grazing pressure.
L377. It is not clear what you mean by previous work emphasizing the role of protective cover. Does this refer to scientific evidence or to speculation and assumptions by previous authors?
L380. Is there just one study implying protective and one study implying obstructive cover as might be inferred from the previous sentence? I think it would be helpful to explain the context of previous research more clearly (i.e. what did they actually find?) Possibly, this should have been done earlier in the manuscript, perhaps in the Introduction, which would allow you to make briefer subsequent references.
L381. I think you mean that it is a limited or incomplete index rather than that it is a partial distance. Even though you explained it previously, it does not make a lot of sense to imply that distance to cover is a poor index of proximity to cover when they are essentially identical words. Revise with more precise wording.
L382. I can see how a poor index could result in a poorer correlation but not how it could reverse the effect of cover. Explain.
L401ff. This section is quite confusing, long, and seems to be quite distant from the topic of your study. I doubt that it contributes substantially to the manuscript. If you feel that there is an important point relevant to the current study, the whole section will need to be carefully revised and condensed (and checked for errors in English). Also 'further than proximity' is a rather ambiguous term. Perhaps you intended it as a play on words, but it does not quite work without a bit more development of what you mean.

---

## Round 0.3 · accepted · Accept

I find the manuscript greatly improved and am happy to recommend it for acceptance. I am glad that the the authors found the comments helpful.